# Novel genes associated with folic acid-mediated metabolism in mouse: A bioinformatics study

**Jianwen Zhao[1], Wen Zou[2], Tingxi Hu[1]\***

**1** Shenyang Medical College, Shenyang, Liaoning, China, **2** Liaoning Vocational College of Ecological Engineering, Shenyang, Liaoning, China

\* htx19871005@126.com

**Data Availability Statement:** All relevant data are within the manuscript and its Supporting Information files.

**Funding:** This research was funded by the Shenyang Medical College Doctoral Research

## Abstract

Folic acid plays an essential role in the central nervous system and cancer. This study aimed to screen genes related to folic acid metabolism. Datasets (GSE80587, GSE65267 and GSE116299) correlated to folic acid were screened in the Gene Expression Omnibus. Weighed gene co-expression network analysis was performed to identify modules associated with sample traits of folic acid and organs (brain, prostate and kidney). Functional enrichment analysis was performed for the eigengenes in modules that were significantly correlated with sample traits. Accordingly, the hub genes and key nodes in the modules were identified using the protein interaction network. A total of 17,252 genes in three datasets were identified. One module, which included 97 genes that were highly correlated with sample traits (including folic acid treatment [cor = -0.57, P = 3e-04] and kidney [cor = -0.68, p = 4e-06]), was screened out. Hub genes, including tetratricopeptide repeat protein 38 (Ttc38) and miR-185, as well as those (including Sema3A, Insl3, Dll1, Msh4 and Snai1) associated with "neuropilin binding", "regulation of reproductive process" and "vitamin D metabolic process", were identified. Genes, including Ttc38, Sema3A, Insl3, Dll1, Msh4 and Snai1, were the novel factors that may be associated with the development of the kidneys and related to folic acid treatment.

## Introduction

Folic acid (vitamin B9), as a necessary micronutrient, has a crucial role in DNA biosynthesis and integrity, controlling the homocysteine level and inflammation response, and reducing the risk of cancers such as colon cancer [1–5]. This also plays an essential role in the central nervous system, and prevents a neural-tube defect (NTD), which is a major birth defect of the brain and spine that occurs early during the embryonic period [3, 6].

Folic acid can reduce the elevated level of serum homocysteine, which is recognized as a risk factor for several diseases, such as cardiovascular and neurological diseases [7]. In addition, it acts as a risk factor for dementia and Alzheimer's disease (AD), and a predictor of cognitive decline [8–10]. In addition to the influence on the central nervous system in the brain,

Startup Fund (No. 20195068), and the Plan of Rejuvenating the Talents of Liaoning (No. XLYC1808012).

**Competing interests:** The authors have declared that no competing interests exist.

plentiful evidence has revealed the influence of folic acid and homocysteine on the health of other organs, such as the kidneys and prostate [11–14]. Significant evidence has indicated that high homocysteine level is associated with increased risk of atherosclerosis, coronary artery disease, and chronic kidney disease (CKD) [13, 14]. The role of folic acid can be well-recognized, because folic acid is an essential cofactor for homocysteine metabolism, and its homeostasis disruption may be directly correlated to cardiovascular risk and CKD progression [15, 16]. A meta-analysis research reported that folic acid therapy can reduce cardiovascular disease risk in patients with CKD by 15% [17]. Cohen et al. [13] conducted a cross-sectional study that involved a large cohort of 17,010 subjects, and concluded that higher homocysteine concentration is correlated with a lower estimated glomerular filtration rate (eGFR) and increased renal impairment. However, the biological mechanism of folic acid in the brain, prostate and kidneys remains unclear, especially in the kidneys. Therefore, it is necessary to distinguish the action mode of folic acid in different organs.

In order to fill this gap, the present bioinformatics study was performed based on the public datasets deposited in online databases. The datasets correlated to folic acid were screened in the National Center for Biotechnology Information (NCBI) Gene Expression Omnibus (GEO) database (http://www.ncbi.nlm.nih.gov/geo/), and the genes related to folic acid metabolism and organ development were identified. The related mechanisms were discussed.

## Materials and methods

### Microarray data resource

Next-generation sequencing datasets related to folic acid were screened in the NCBI GEO database (http://www.ncbi.nlm.nih.gov/geo/) using the following search strategy: 'folic acid' OR 'folate' AND 'Mus musculus [Organism]'. Three datasets, including GSE80587, GSE65267 and GSE116299, were selected. GSE80587 (GPL13112, Illumina HiSeq 2000) consisted of 11 samples isolated from the hippocampi of six control mice fed with control dietary (14-week old, F1; three males and three females) and five mice fed with folic acid dietary at 40 mg/kg of chow (14-week old, F1, two males and three females) [18]. GSE65267 (GPL13112, Illumina HiSeq 2000) included 18 samples isolated from mouse kidney tissues before and at various time points (1, 2, 3, 7 and 14 days) after a single intraperitoneal injection of folic acid (250 mg/kg, n = 3/time-point) [19]. GSE116299 (GPL21493, Illumina HiSeq 3000) comprised of 23 samples isolated from prostate tissues from intact or castrated male mice (3, 10 and 14 days post-castration) fed with a control (4 mg of folic acid/kg of feed) and folic acid supplemented diet (24 mg of folic acid/kg of feed) from conception [20]. The data from the prostate tissues of intact male mice (n = 4/group) were used in the present study. The data files were downloaded from the GEO (GSE80587 and GSE65267) or European Nucleotide Archive (ENA) database (GSE116299) in the European Bioinformatics Institute (EBI; https://www.ebi.ac.uk/ena).

### Data preprocessing

The raw data in GSE80587 and GSE65267 were processed using the Limma package (version 3.10.3, http://www.bioconductor.org/packages/2.9/bioc/html/limma.html) in R [21]. The workflow is shown in S1 Fig. The data (fastq) downloaded from GSE116299 was processed using fastQC to filter the low quality data [22]. The HISAT2 software (version 2.1.0; http://ccb.jhu.edu/software/hisat2) [23] was used to align to the reference genome of the mouse (mm10), with the default parameters. The raw counts of genes were calculated using featurecounts (Version 1.6.0) [24], and the transcripts per million (TPM) value of each gene was calculated for each sample. The Limma package was used for the batch effect correction [25]. Genes with averaged TPM from duplicates over 0.1 [26] and the coefficient of variation that ranked the

top 75% were reserved [27]. According to the TPM values, the sample cluster diagram was drawn, and the outliers were removed.

## Selection of gene modules through weighed gene co-expression network analysis (WGCNA)

The weighed gene co-expression network analysis (WGCNA) is usually applied for integrating the gene expression and identifying modules that are associated with sample traits [28]. All samples were pooled and assigned into two groups, according to the treatment strategy (with and without folic acid) or organ (brain, kidneys and prostate). The modules related to the organ and folic acid were analyzed using the WGCNA package in R (version 1.61, https://cran.r-project.org/web/packages/WGCNA/), according to the scale-free network theory [29]. Stable gene modules were screened according to the TPM values of all genes, with the thresholds of minModuleSize = 30, dissimilarity = 0.25, softPower = 11, and cutHeight = 0.95. The module significance criteria were set as P<0.05, and the correlation coefficient (cor) was >0.5. The eigengenes in the selected modules were used for further analysis.

## Functional enrichment analysis

In order to investigate the functional processes and pathways associated with the eigengenes in the significant modules, the functional enrichment analysis was separately conducted for the eigengenes in each module. Database for Annotation, Visualization, and Integrated Discovery (DAVID) online tool (version 6.8, https://david.ncifcrf.gov/) [30] was utilized to extract the meaningful Gene Ontology (GO) functional terms, including the biological processes (BP), molecular function (MF) and cellular components (CC), and the Kyoto Encyclopedia of Genes and Genomes (KEGG) pathways. Significant items were selected according to the criteria of P-value <0.05.

## Protein-protein interaction (PPI) network construction

First, the eigengenes in the significant modules were subjected to the STRING (Version 10.0, http://string-db.org/) database, and the protein interaction pairs (score >0.4) were extracted. The PPI network was separately constructed for eigengenes in each module. The PPI network of each significant module was constructed using the Cytoscape software (Version 3.2.0, http://www.cytoscape.org/) [31]. The genes for coding the nodes in the PPI networks were regarded as hub genes.

# Results

## Data processing and batch effect correction

After data processing, a total of 17,252 genes with TPM >0.1 were identified from the samples in the three datasets (GSE80587, GSE65267 and GSE116299). The Pearson's correlation analysis revealed that samples had a high correlation intra-dataset (Fig 1A). The batch effect correction for the 37 samples from three datasets indicated an outlier in GSE65267, which was named as, FA3dayrep1 (Fig 1B). The sample FA3dayrep1 was removed, accordingly.

## WGCNA module selection

Before the WGCNA, the soft-thresholding power of the adjacency matrix was explored, according to the scale-free network theory. The value of soft-thresholding power was 11, when the square value of the correlation coefficient ($r^2$) = 0.9 (Fig 2A). The mean connectivity = 1 when the soft-thresholding power was 11 (Fig 2B). According to the parameters and criteria

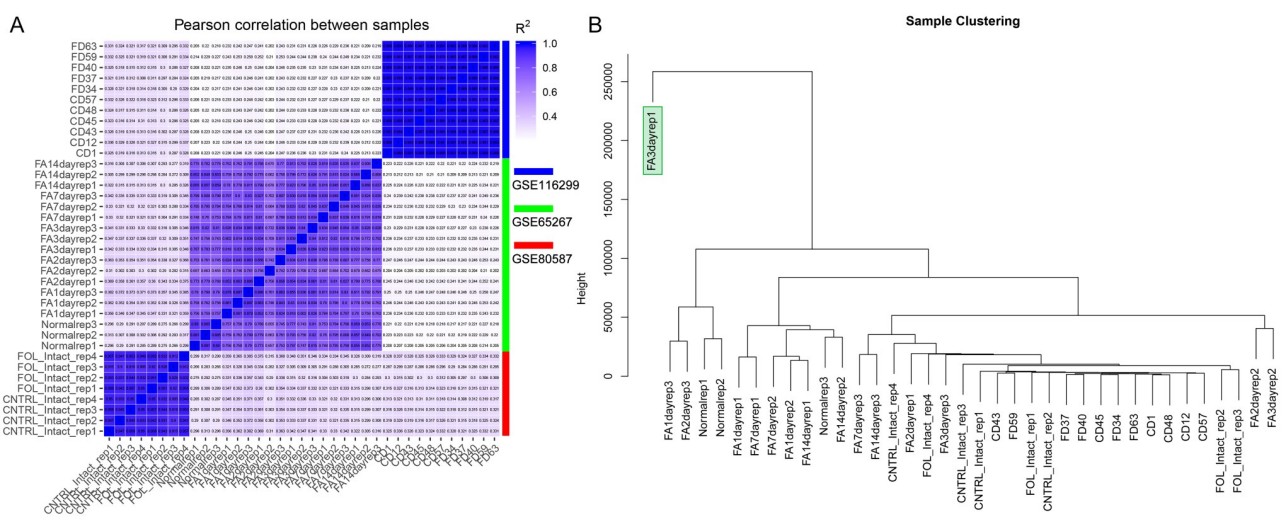

**Fig 1. Sample correlation and batch effect correction.** (A) The Pearson's correlation analysis for the samples included in the GSE80587, GSE65267 and GSE116299 datasets. (B) The batch effect correction of samples in the GSE80587, GSE65267 and GSE116299 datasets.

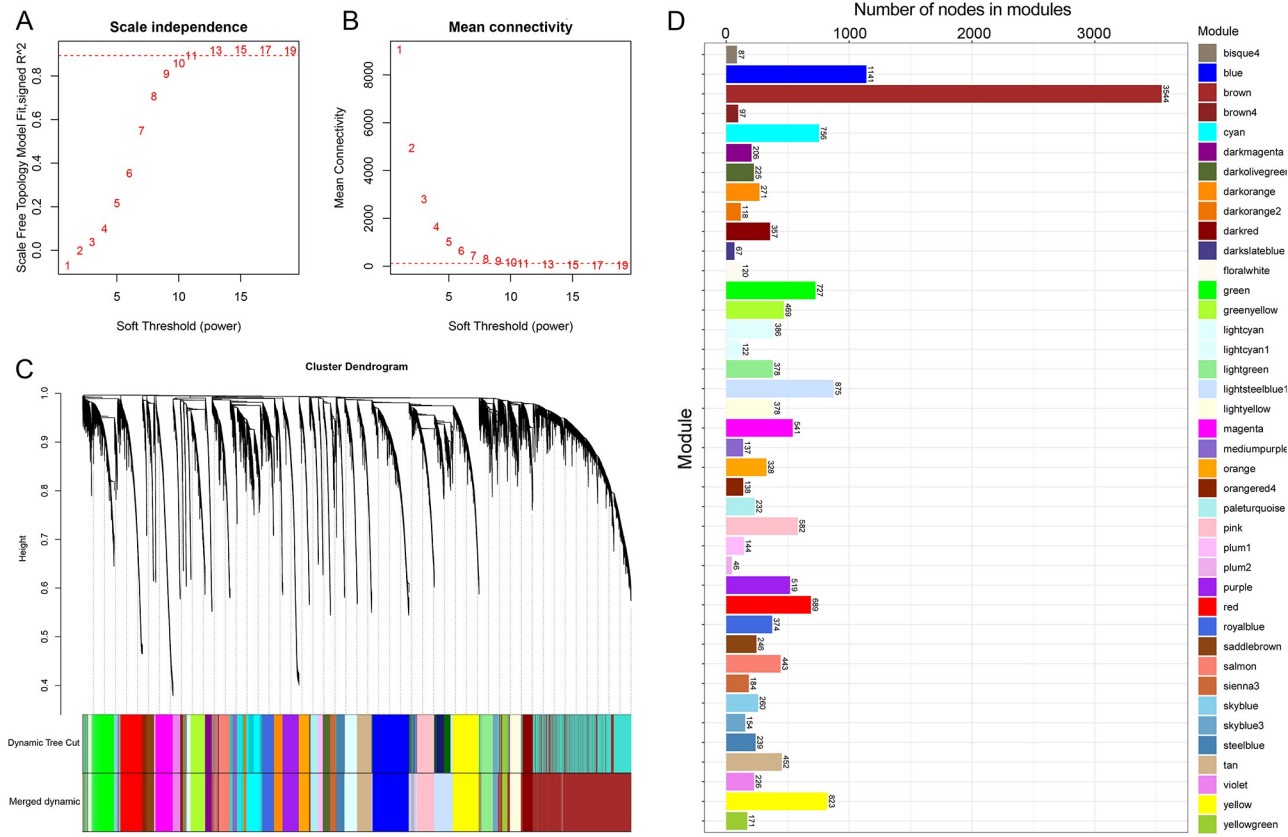

**Fig 2. The network topology analysis for the soft threshold power of the adjacency matrix.** (A) The soft-thresholding power corresponding to the correlation coefficient square value ($r^2$, y-axis). The higher the $r^2$ value, the closer this was to the scale-free topology. (B) The connectivity corresponding to the different soft-thresholding power. The higher the soft-thresholding power, the lower the mean connectivity. The mean connectivity was equal to 1 when soft-thresholding power = 11. (C) The WGCNA modules tree. (D) The WGCNA modules and number of nodes in the modules.

**Module−trait relationships**

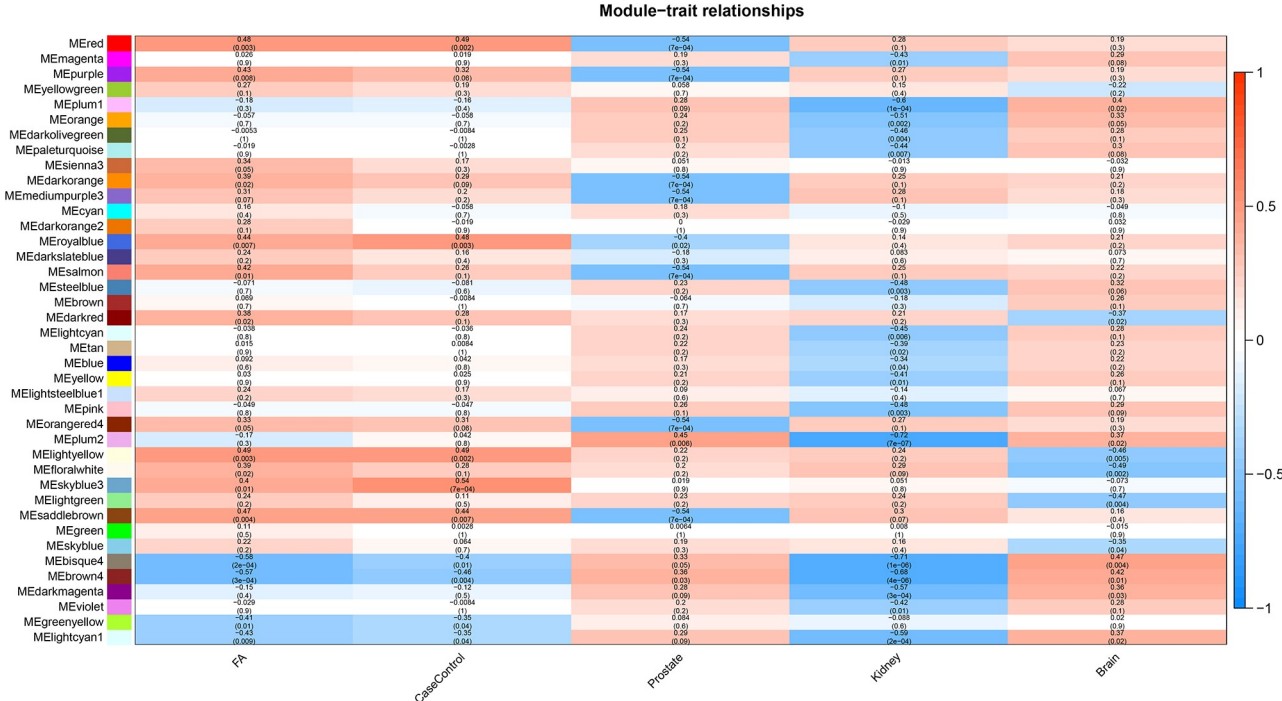

**Fig 3. The correlation of modules with the sample trait.** ME, module of eigengenes. Red and blue colors in the heatmap note the positive and negative correlations with the corresponding traits, respectively. *Indicates the significant correlation (absolute cor > 0.5 and P<0.05).

mentioned above (minModuleSize = 30, dissimilarity = 0.25, softPower = 11, and cutHeight = 0.95), the module tree was constructed (Fig 2C), and a total of 40 co-expressed modules were identified (Fig 2D).

These modules consisted of 46–3,544 eigengenes, respectively. In addition, 2, 7 and 7 modules were correlated to the folic acid treatment, prostate and kidneys, respectively (Fig 3). It was noted that two modules (bisque4, gene count = 87; and brown4, gene count = 97) were significantly correlated with the folic acid treatment (cor = -0.58, p = 2e-04 and cor = -0.57, p = 3e-04, respectively; Fig 3) and the kidney trait (cor = -0.71, p = 1e-06 and cor = -0.68, p = 4e-06, respectively).

The sample expression profiles of the eigengenes in bisque4 and brown4 are shown in S1 Fig. The correlation analysis revealed that the 97 eigengenes in the brown4 module revealed significant and moderate correlation (cor = 0.51 and $P < 9.5e-08$, S2A Fig), while the 87 genes in bisque4 had a low correlation (cor = 0.40 and $P = 0.00012$, S2B Fig). Then, the brown4 module was identified as the key one in the present study.

## Enrichment analysis

The GO functional analysis revealed that eigengenes in the brown4 module were associated with BPs such as "regulation of male gonad development" (including Semaphorin 3A, Sema3A; and insulin-like factor 3, Insl3), "cell surface receptor signaling pathway involved in heart development" (involves Snai1 and Dll1), "regulation of vitamin metabolic process", and "vitamin D metabolic process" (Snai1; Fig 4A and S1 Table). Sema3A had the MFs of "semaphorin receptor binding", "chemorepellent activity", and "neuropilin binding", and Dll1 acted its activity "Notch binding" as a component of "apical part of cell" (S1 Table). Sema3a, MutS homologue 4 (Msh4), and Snai1 genes were involved in "regulation of reproductive process",

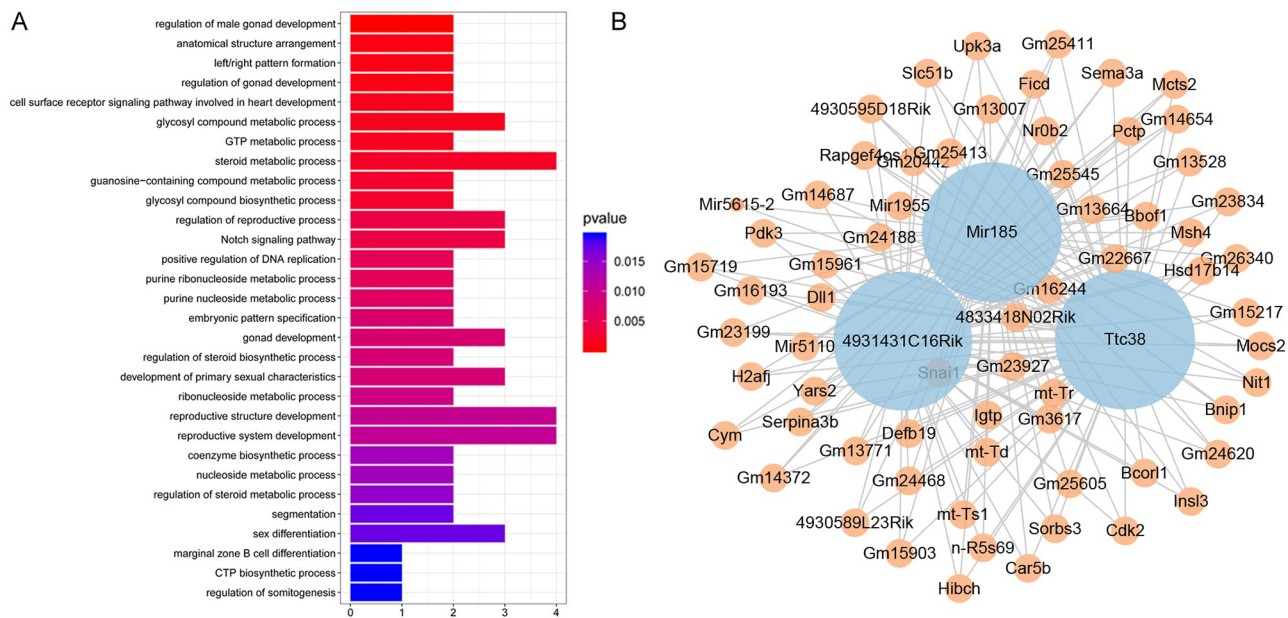

**Fig 4. Functional enrichment analysis and the protein-protein interaction (PPI) network of eigengenes in the brown4 module.** (A) The GO biological processes that involve the eigengenes in the brown4 module. (B) The PPI network of eigengenes in the brown4 module. The node size indicates the interaction degree.

"gonad development", "reproductive system development", "female gamete generation", and "sex differentiation" (S1 Table).

## PPI network and hub genes

The PPI network of the eigengenes in the brown4 module consisted of 69 genes, including eight hub genes with Kmer >0.95, including tetratricopeptide repeat protein 38 (Ttc38), miR-185 and 4931431C16Rik, with the interaction degree of 68, 68 and 67 in the network, respectively (Fig 4B). The other five hub genes were solute transporter-β (Slc51b), Msh4, Gm14687, Gm23199, n-R5s69 and Gm20442, with a low interaction degree in the PPI network (degree = 3).

## Discussion

This study identified that a WGCNA module (brown4) and key hub genes were simultaneously associated with folic acid-related mechanisms in the kidney. Among the eigengenes in brown4, one hub gene Ttc38 and one key miRNA miR-185 were identified. Eigengenes in the brown4 module, including Sema3A, Insl3, Dll1, Msh4 and Snai1, were associated with "regulation of reproductive process", "gonad development", "sex differentiation", "neuropilin binding" and "regulation of vitamin metabolic process". These nodes are the novel factors associated with folic acid metabolism or kidney development.

Folic acid deficiency correlates with elevated homocysteine levels, which is a risk factor for colon cancer [1, 32]. Larriba et al. [33] reported that colon cancer tissues that co-express Snai2 and Snai1 downregulated the vitamin D receptor, which mediates the antitumoral action of vitamin D. Sema3A is a ligand of neuropilin-1 and a tumor suppressor in acute leukemia [34]. Neuropilin-1 is involved in diverse processes, including cancer and angiogenesis [35–37]. It is a transmembrane glycoprotein that is required for the development of embryonic neuron and

vascular [37, 38]. Elevated neuropilin-1 in urinary and renal tissues is associated with the clinical response of renal lupus nephritis [35]. The present study identified that Sema3A, Insl3, Dll1, Msh4 and Snai1 were all negatively associated with folic acid treatment and kidney development, which reveals their potential roles in folic acid-mediated diverse functions.

Homocysteine is a product of the methylation cycle and is catalyzed to methionine by enzyme methionine synthetases (MSs) [39]. Methylation reactions involve almost all chemical reactions in body, and its disturbance has been linked to various body disorders, including brain atrophy, oxidative stress, increased apoptosis, DNA damage and neurodegenerative disorders, such as Parkinson's disease, AD and depression [8, 40]. Increasing evidence has shown folic acid and vitamin $B_{12}$ can significantly improve cognitive performance in patients with AD [41, 42]. A 5-year trial for a large cohort of postmenopausal women without symptoms of dementia (memory cognitive impairment) indicated that lower levels of folic acid than the recommended daily allowance ($<$400 µg/d) increased the risk of dementia and cognitive impairment [43]. The benefit of folic acid in the central nervous system is attributed to its effect on homocysteine [8–10]. Folic acid can markedly increase the serum S-adenosylmethionine (SAM), which is a key MS that catalyzes methylation reactions in cells [4, 44], and thereby declines the accumulation of homocysteine and mediates cytotoxicity, DNA damage and neurodegenerative disorders.

Homocysteine, which can induce cytotoxicity, apoptosis and autophagy, is correlated to various cell signaling pathways, such as phosphoinositide 3-kinase (PI3K)/Akt [45–47]. Liu et al. [46] conducted an *in vitro* study by treating human umbilical vein endothelial cells with homocysteine, with and without epigallocatechin gallate, and this prevented homocysteine-induced cell apoptosis by activating he PI3K/Akt/endothelial nitric oxide synthase (eNOS) pathway. Price et al [11] reported that higher folate concentration was associated with elevated risk of prostate cancer (95% confidence interval [CI], 1.02–1.26) and high-grade disease. Prostate-specific membrane antigen (PSMA) or folate hydrolase 1 (FOLH1) is overexpressed in prostate cancer, and correlates with the PI3K/Akt signaling in cells [48, 49]. The inhibition of PSMA conversely promotes tumor regression by inhibiting PI3K signaling in preclinical models [12]. MiR-185 acts as a tumor suppressor, and inhibits tumor progression by regulating its targets, including the Akt1 and PI3K/AKT pathway expression [50]. The present study revealed that Ttc38 is a target of miR-185. Ttc38 was included in the brwon4 module, which is associated with folic acid or the kidneys, while the direct association of Ttc38 with these was not identified via bioinformatics analysis. These data revealed that Ttc38 is a novel factor that may be associated with folic acid or kidney development.

High homocysteine level is associated with increased risk of chronic kidney disease [13, 14]. The present study revealed that the Ttc38 associated with folic acid was also negatively correlated with the kidney. Another tetratricopeptide repeat (TPR) member, Ttc36, has an organ-specific expression profile, and shows a high expression level in the kidneys and liver, and a low expression level in the testis [51]. The expression of Ttc36 in renal proximal tubules is spatially and temporally-specific [51]. A recent proteome of mouse with experimental autoimmune encephalomyelitis (EAE) identified the downregulation of Ttc38 in the brain [52]. The present study revealed that Ttc38 is kidney-specific and folic acid-related. The novel expression profile may suggest the interesting mechanism in the folic acid-related mechanism, which may be kidney-specific.

These findings may also show the relationship between kidney disease and folic acid. Xu reported that enalapril-folic acid therapy, compared with enalapril alone, can significantly delay the progression of chronic kidney disease (CKD) in patients with mild-to-moderate CKD [53]. In addition, Matsumoto et al. reported that treatments with folic acid in phase G3b and G4 may reduce renal disease progression by enhancing antioxidant defenses [54]. These

evidences further support that folic acid is correlated to the kidneys. The expression levels of these candidate genes may be regulated by folic acid, which in turn participates in kidney functions. Although no further cellular or animal experiments were carried out on candidate genes in this study, the present study can provide new insights for kidney development and kidney disease research.

## Conclusion

In conclusion, the present study revealed a WGCNA module that consisted of genes associated with folic acid and the kidneys. Hub genes, such as Ttc38 and miR-185, are key factors that have a significant relationship with the development of the kidneys after folic acid treatment. In addition, genes such as Sema3A, Insl3, Dll1, Msh4 and Snai1 may have potential roles in regulating metabolisms correlated to folic acid and kidney development. These evidences further support that folic acid is correlated to the kidneys. The expression levels of these candidate genes may be regulated by folic acid, which in turn participates in kidney function.

## Supporting information

**S1 Table. The GO functional enrichment analysis of the eigengenes in the brown4 module.**
(XLSX)

**S1 Fig. Heatmap of the expression of eigengenes in the bisque4 (A) and brown4 (B) module, respectively.**
(TIF)

**S2 Fig. Gene significance of eigengenes in the brown4 (A) and bisque4 (B) module, respectively.**
(TIF)

## Author Contributions

**Data curation:** Jianwen Zhao, Tingxi Hu.

**Formal analysis:** Tingxi Hu.

**Methodology:** Wen Zou.

**Writing – original draft:** Tingxi Hu.

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
