## [Decision Letter · Decision Letter 0]

27 May 2020

PONE-D-20-10567

Novel genes associates with folic acid-mediated metabolism in mouse: a bioinformatics study

PLOS ONE

Dear Dr. Hu,

Thank you for submitting your manuscript to PLOS ONE. After careful consideration, we feel that it has merit but does not fully meet PLOS ONE’s publication criteria as it currently stands. Therefore, we invite you to submit a revised version of the manuscript that addresses the points raised during the review process.

We would be willing to consider a revised manuscript that addresses all the comments raised by the reviewers. A key concern is the quality of presentation (especially for the figures); the manuscript should be revised to address the relevant issues raised by both reviewers. Additionally, there are many grammatical and typographical errors throughout the manuscript that should be fixed.

We look forward to receiving your revised manuscript.

Kind regards,

Jishnu Das, Ph.D.

Academic Editor

PLOS ONE

Journal Requirements:

Reviewers' comments:

Reviewer's Responses to Questions

**Comments to the Author**

1. Is the manuscript technically sound, and do the data support the conclusions?

Reviewer #1: Yes

Reviewer #2: Yes

2. Has the statistical analysis been performed appropriately and rigorously? 

Reviewer #1: Yes

Reviewer #2: I Don't Know

3. Have the authors made all data underlying the findings in their manuscript fully available?

Reviewer #1: Yes

Reviewer #2: Yes

4. Is the manuscript presented in an intelligible fashion and written in standard English?

Reviewer #1: Yes

Reviewer #2: No

5. Review Comments to the Author

Reviewer #1: This study aims to investigate genes associated with folic acid metabolism. Folic acid levels have been associated with different health indicators in the brain, prostate, and kidneys, but the mechanism by which folic acid influences these tissues is unclear. A computational analysis of genes in the tissue may help shed light on key gene networks in these processes. The authors take an informatic approach by investigating public NGS data from the NCBI GEO database. They specifically target mouse experiments involving folic acid for their data, and subsequently perform weighted gene coexpression network analysis (WGCNA), functional enrichment analysis, and protein-protein interaction network construction. After analysis, they found several novel genes have been connected to folic acid metabolism and renal development.

The general structure of the research and the paper is good. The authors explicitly state their objective to identify genes associated with folic acid metabolism. They do a good job of identifying previous studies correlating folic acid with biomarkers in different contexts. They then contextualize their study as an investigation towards the underlying molecular processes behind these associations. For the purpose of their exploratory analysis, the methods were suitable to address the research question. Their design flowed well in using the findings of one analysis in the one immediately after, so the rationale was clear altogether. The methods for the analyses also referenced literature sufficiently in justifying their use of statistical packages/software for a task. However, some parts of the methods were a little ambiguous. The biggest issue was in the figures. Some presentation decisions made the figures very messy and distracted from their significance. Furthermore, the captions could use more detail in explaining the value of various parts of a figure. Additionally, the writing throughout the paper was dotted with spacing issues, misspellings, and grammatical errors. In some cases, these created comprehension issues that were only cleared up after a full readthrough. Ultimately, the presented conclusions were supported by the results in the paper. Overall, I found that the quality of research is appropriate for publication in PLOS ONE, but its presentation in terms of writing and figures should be refined further.

Major:

• Some of the figures require further refinement. For Figure 1A, the numbers in the boxes are pixelated even in the high-resolution representation, and it is unclear how much the value they add in comparison to a separate supplemental table. Figure 4B is very messy overall. Shortening the names for the nodes or omitting some altogether would significantly improve legibility here. Figure S2 seems unclear for interpretation. Adding a regression line may help.

• For figure 2, it is unclear why the soft power of 11 was being highlighted in 2A and 2B when a soft power of 1 was ultimately chosen for the WGCNA analysis shown in 2C. For Figure 2C, more detail is required, as the figure doesn’t clearly connect to the authors’ claims that 40 modules were identified. Detailing the caption should help clarify this figure.

• I couldn’t find any explicit discussions of study limitations or potential future avenues for investigation, so acknowledging some issues could significantly strengthen the conclusion and help contextualize the research.

• The writing should be improved significantly to help the paper flow more smoothly.

Minor:

• In their introduction, the authors devote a lot of time to the central nervous system, a few lines to the kidneys, and very little to the prostate. While the current organization may reflect the state of folic acid research, the introduction should be rebalanced in the context of the paper’s findings. More focus on kidneys, a little more on prostate, and less on the CNS.

• Citing literature justification explicitly for the TPM cutoff and coefficient of variation would further strengthen the paper.

• The small amount of mice (<100 in total) could be increased to lend validity to the findings, especially since the mice are draw from 3 separate studies.

• Publishing code may help reduce or expose any concerns with study reproducibility.

• Conducting more parallel statistical analyses or performing experimental validation may help strengthen findings and research quality.

Reviewer #2: This manuscript describes the study for the identification of the genes related to folic acid metabolism and organ development by applying a bioinformatics pipeline based on public datasets. This study may be of interest to the readers, however, I have some recommendations for the authors.

Can the authors perform a sensitivity analysis or at least comment on how much the results are sensitive to the choice of parameter values or settings of the algorithms or packages that have been used in the bioinformatics pipeline?

The structure of the manuscript can be improved to make it easier to follow. For example, the flow of the bioinformatics pipeline can be given either in text format or by a figure. Also, Figure 4 can be redrawn since there is too much text in its current form.

Also, there are some typographical and spelling errors in the manuscript. Some examples are indicated below.

1. Material and Methods pp.8, line 212

“(fastq) downloaded from GSE116299 was processed using fastQ for filtering the low quality”

I suppose fastQ should be corrected ad fastQC and I think that the reference number 39 is not the the correct reference for fastQC

The correct citation can be

Andrews, S. (2010). FastQC: A Quality Control Tool for High Throughput Sequence Data [Online]. Available online at: http://www.bioinformatics.babraham.ac.uk/projects/fastqc/

2. Correlationintra written as a single word

pp.3-line 70 analysis showed samples had high correlationintra-dataset (Figure 1A).

3. softPower=1 should be corrected as softPower=11

pp.3-line 82 when soft-thresholding power was 11 (Figure 2B). According to the parameters and criteria

83 of mentioned above (minModuleSize=30, dissimilarity=0.25, softPower=1 and

4. showedsiginificant written as a single word

pp.4-line 104 S1. Correlation analysis showed that the 97eigengenes in brown4 module showedsignificant

5. bisque4showed written as a single word

pp.4-line 106 bisque4showed low correlation (cor=0.40 and P=0.00012, Figure S2B). Then, the brown4

6. PLOS authors have the option to publish the peer review history of their article (what does this mean?). If published, this will include your full peer review and any attached files.

Reviewer #1: No

Reviewer #2: Yes: Volkan Atalay

---

## [Author Response · Author response to Decision Letter 0]

29 Jun 2020

Dear editor,

Thank you for giving us the opportunity to revise the manuscript. We gladly accept the reviewers’ comments and revise them one by one. We hope that this revised manuscript will meet the publishing requirements.

The point by point response were listed below.

Thanks for the editor's work, and the reviewers for their careful review.

Kind regards,

Tingxi Hu

Reviewer #1: This study aims to investigate genes associated with folic acid metabolism. Folic acid levels have been associated with different health indicators in the brain, prostate, and kidneys, but the mechanism by which folic acid influences these tissues is unclear. A computational analysis of genes in the tissue may help shed light on key gene networks in these processes. The authors take an informatic approach by investigating public NGS data from the NCBI GEO database. They specifically target mouse experiments involving folic acid for their data, and subsequently perform weighted gene coexpression network analysis (WGCNA), functional enrichment analysis, and protein-protein interaction network construction. After analysis, they found several novel genes have been connected to folic acid metabolism and renal development.

The general structure of the research and the paper is good. The authors explicitly state their objective to identify genes associated with folic acid metabolism. They do a good job of identifying previous studies correlating folic acid with biomarkers in different contexts. They then contextualize their study as an investigation towards the underlying molecular processes behind these associations. For the purpose of their exploratory analysis, the methods were suitable to address the research question. Their design flowed well in using the findings of one analysis in the one immediately after, so the rationale was clear altogether. The methods for the analyses also referenced literature sufficiently in justifying their use of statistical packages/software for a task. However, some parts of the methods were a little ambiguous. The biggest issue was in the figures. Some presentation decisions made the figures very messy and distracted from their significance. Furthermore, the captions could use more detail in explaining the value of various parts of a figure. Additionally, the writing throughout the paper was dotted with spacing issues, misspellings, and grammatical errors. In some cases, these created comprehension issues that were only cleared up after a full readthrough. Ultimately, the presented conclusions were supported by the results in the paper. Overall, I found that the quality of research is appropriate for publication in PLOS ONE, but its presentation in terms of writing and figures should be refined further.

Response: Thanks for your review, we are willing to accept your suggestions and do our best to revise the article.

Major:

• Some of the figures require further refinement. For Figure 1A, the numbers in the boxes are pixelated even in the high-resolution representation, and it is unclear how much the value they add in comparison to a separate supplemental table. Figure 4B is very messy overall. Shortening the names for the nodes or omitting some altogether would significantly improve legibility here. Figure S2 seems unclear for interpretation. Adding a regression line may help.

Response: Thanks for your comment. We have revised all the figures to improve the quality and readability. Furthermore,more details were provided in the captions part.

• For figure 2, it is unclear why the soft power of 11 was being highlighted in 2A and 2B when a soft power of 1 was ultimately chosen for the WGCNA analysis shown in 2C. For Figure 2C, more detail is required, as the figure doesn’t clearly connect to the authors’ claims that 40 modules were identified. Detailing the caption should help clarify this figure.

Response: Thanks for your comment.We are sorry for incorrectly marking the value of soft power in the text. Figure 2C is analyzed under the parameters of 11, not 1. We have revised it in the manuscript.Figure 2C does indeed mislead the reader, so we added nodes statistical results for modules (Figure 2D). in addition, the captions were updated to clarify this figure.

• I couldn’t find any explicit discussions of study limitations or potential future avenues for investigation, so acknowledging some issues could significantly strengthen the conclusion and help contextualize the research.

Response: Thanks for your comment. We added both the limitations and potential future avenues in this manuscript. Our findings may remind us of the relationship between kidney disease and folic acid. Xu found that enalapril-folic acid therapy, compared with enalapril alone, can significantly delay the progression of chronic kidney disease (CKD) among patients with mild-to-moderate CKD. Also, Matsumoto et al found that treatment with folic acid in phase G3b and G4 may reduce renal disease progression by enhancing antioxidant defenses. These evidences further proved that folic acid was related to the kidney. The expression levels of these candidate genes may be regulated by folic acid, which in turn participates in kidney function. Although no further cellular or animal experiments have been carried out on candidate genes in this study, this present study can provide new insight for kidney development and kidney disease research.

• The writing should be improved significantly to help the paper flow more smoothly.

Response: Thanks for your comment.We improved the language by AJESCI language retouching service.

Minor:

• In their introduction, the authors devote a lot of time to the central nervous system, a few lines to the kidneys, and very little to the prostate. While the current organization may reflect the state of folic acid research, the introduction should be rebalanced in the context of the paper’s findings. More focus on kidneys, a little more on prostate, and less on the CNS.

Response: Thanks for your comment.We made major revisions to the introduction, reduced the description of AD. We more focus on kidneys and CDK.

• Citing literature justification explicitly for the TPM cutoff and coefficient of variation would further strengthen the paper.

Response: Thanks for your comment. The literatures were cited for the TPM cutoff and coefficient of variation.Kagale et al used a similar similarTPM cutoff parameter in their WGCNA analysis. Liao reported the method of the coefficient of variation range.

• The small amount of mice (<100 in total) could be increased to lend validity to the findings, especially since the mice are draw from 3 separate studies.

Response: Thank you for your suggestion. We agree with your comment very much. We consulted many literatures and removed some of them according to the research background, and found that not many data were available. So, the number of mice does not reach 100. In addition, the number of mice used in this article is in accordance with WGCNA requirements (https://horvath.genetics.ucla.edu/html/CoexpressionNetwork/Rpackages/WGCNA/faq.html), so the results of this study are credible.

• Publishing code may help reduce or expose any concerns with study reproducibility.

Response: Thank you for your suggestion.We are very sorry that the code we used is an in-housepipeline, so it cannot be provided. But the manuscript already described all the parameters used.

• Conducting more parallel statistical analyses or performing experimental validation may help strengthen findings and research quality.

Response: Thank you for your suggestion.This study is mainly to reuse the published data by bioinformatic analysis, and then to screen the genes that related to folic acid treatment in kidney.The verification work is necessary, but it is also a pity that we are limited by factors such as animal materials.Therefore, we also discussed the limitations of this study in the discussion section.

Reviewer #2:This manuscript describes the study for the identification of the genes related to folic acid metabolism and organ development by applying a bioinformatics pipeline based on public datasets. This study may be of interest to the readers, however, I have some recommendations for the authors.

Can the authors perform a sensitivity analysis or at least comment on how much the results are sensitive to the choice of parameter values or settings of the algorithms or packages that have been used in the bioinformatics pipeline?

Response: Thank you for your comment. Part of parameters in our WGCNA analysis were choose base on the value that most papers published or the WGCNA official default setting (https://horvath.genetics.ucla.edu/html/CoexpressionNetwork/Rpackages/WGCNA/Tutorials/index.html), so the result is robust and repeatable.Thearticles listed below used the similar parameters of WGCNA.

Young C D, Dammer E, Griffen T, et al. WGCNA identification of CXCL13 and associated genes involved in the Tumor Immune Microenvironment (TIME) of lung adenocarcinoma[J]. 2020.

Di Y, Chen D, Yu W, et al. Bladder cancer stage-associated hub genes revealed by WGCNA co-expression network analysis[J]. Hereditas, 2019, 156(1): 7.

Feltrin A S A, Tahira A C, Simoes S N, et al. Assessment of complementarity of WGCNA and NERI results for identification of modules associated to schizophrenia spectrum disorders[J]. PloS one, 2019, 14(1).

Kagale S, Nixon J, Khedikar Y, Pasha A, Provart NJ, Clarke WE, et al. The developmental transcriptome atlas of the biofuel crop Camelina sativa. The Plant Journal. 2016; 88(5): 879-894. https://doi.org/10.1111/tpj.13302.

Liao E. Challenges in High-throughput Data Analysis: Proteomic Data Pre-processing and Network Methods for Integrating Multiple Data Types: UCLA; 2012

The structure of the manuscript can be improved to make it easier to follow. For example, the flow of the bioinformatics pipeline can be given either in text format or by a figure. Also, Figure 4 can be redrawn since there is too much text in its current form.

Response: Thank you for your comment.We have restructured the article structure according to the author's instructions. The flow of the bioinformatics pipeline is shown in Figure S1. In addition, Figure 4 was redrawn. We rename the nodes to gene symbles.

Also, there are some typographical and spelling errors in the manuscript. Some examples are indicated below.

1. Material and Methods pp.8, line 212

“(fastq) downloaded from GSE116299 was processed using fastQ for filtering the low quality”

I suppose fastQ should be corrected ad fastQC and I think that the reference number 39 is not the the correct reference for fastQC

The correct citation can be

Andrews, S. (2010). FastQC: A Quality Control Tool for High Throughput Sequence Data [Online]. Available online at: http://www.bioinformatics.babraham.ac.uk/projects/fastqc/

Response: Thank you for your comment. We revised it according to your comment.

2. Correlationintra written as a single word

pp.3-line 70 analysis showed samples had high correlationintra-dataset (Figure 1A).

Response: Thank you for your comment. We revised it according to your comment.

3. softPower=1 should be corrected as softPower=11

pp.3-line 82 when soft-thresholding power was 11 (Figure 2B). According to the parameters and criteria

83 of mentioned above (minModuleSize=30, dissimilarity=0.25, softPower=1 and

Response: Thank you for your comment. the parameter is 11, not 1. Revised it as required.

4. showedsiginificant written as a single word

pp.4-line 104 S1. Correlation analysis showed that the 97eigengenes in brown4 module showedsignificant

Response: Thank you for your comment. Revised it as required.

5. bisque4showed written as a single word

pp.4-line 106 bisque4showed low correlation (cor=0.40 and P=0.00012, Figure S2B). Then, the brown4

Response: Thank you for your comment. Revised it as required.

---

## [Decision Letter · Decision Letter 1]

21 Jul 2020

PONE-D-20-10567R1

Novel genes associate with folic acid-mediated metabolism in mouse: a bioinformatics study

PLOS ONE

Dear Dr. Hu,

Thank you for submitting your manuscript to PLOS ONE. After careful consideration, we feel that it has merit but does not fully meet PLOS ONE’s publication criteria as it currently stands. Therefore, we invite you to submit a revised version of the manuscript that addresses the points raised during the review process.

ACADEMIC EDITOR:

The reviewers raised a few additional concerns that should be addressed. The manuscript would also improve from further proofreading and language editing.

We look forward to receiving your revised manuscript.

Kind regards,

Jishnu Das, Ph.D.

Academic Editor

PLOS ONE

Reviewers' comments:

Reviewer's Responses to Questions

**Comments to the Author**

1. If the authors have adequately addressed your comments raised in a previous round of review and you feel that this manuscript is now acceptable for publication, you may indicate that here to bypass the “Comments to the Author” section, enter your conflict of interest statement in the “Confidential to Editor” section, and submit your "Accept" recommendation.

Reviewer #1: All comments have been addressed

Reviewer #2: All comments have been addressed

2. Is the manuscript technically sound, and do the data support the conclusions?

Reviewer #1: Yes

Reviewer #2: Yes

3. Has the statistical analysis been performed appropriately and rigorously? 

Reviewer #1: Yes

Reviewer #2: Yes

4. Have the authors made all data underlying the findings in their manuscript fully available?

Reviewer #1: Yes

Reviewer #2: Yes

5. Is the manuscript presented in an intelligible fashion and written in standard English?

Reviewer #1: Yes

Reviewer #2: Yes

6. Review Comments to the Author

Reviewer #1: The authors did a good job of addressing overarching concerns as well as specific prior issues raised by modifying the paper and directly describing the changes in response to the comments. Overall, I have no other significant concerns, but I did notice a few final typos to fix.

Final revisions:

• Line 34: “HubgenesHub genes” should be changed to just “Hubgenes” or “Hub genes”

• Line 84: “10and 14” should be fixed to “10 and 14”

• Line 162: “wasnoted” should be broken up into “was noted”

• 172: “97eigengenes” and “showedsignificant” should be split into “97 eigengenes” and “showed significant”

• 195: “Hubgeneswith” should be split up

• 223: “B12can” should be split up

• Figure 2D should be retitled “Number of nodes in modules” or something along those lines. Numbers is currently misspelled as numbers.

Reviewer #2: The authors addressed my comments. However, there are still some typographical errors.

These errors are all two words written together and they can be corrected easily by using a spellchecker.

pp1 line 34 HubgenesHub genes

pp6 line 195 hubgeneswith

pp6 line 200 weresimultaneously

pp6 line 202 wereidentified

pp7 line 241 promotestumor

pp8 line 273 mayhave

7. PLOS authors have the option to publish the peer review history of their article (what does this mean?). If published, this will include your full peer review and any attached files.

Reviewer #1: No

Reviewer #2: No

---

## [Author Response · Author response to Decision Letter 1]

27 Jul 2020

Dear editor,

Thank you for giving us the opportunity to revise the manuscript. We gladly accept the reviewers’ comments and revise them one by one. We hope that this revised manuscript will meet the publishing requirements.

The point by point response were listed below.

Thanks for the editor's work, and the reviewers for their careful review.

Kind regards,

Tingxi Hu

Reviewer #1: The authors did a good job of addressing overarching concerns as well as specific prior issues raised by modifying the paper and directly describing the changes in response to the comments. Overall, I have no other significant concerns, but I did notice a few final typos to fix.

Final revisions:

• Line 34: “HubgenesHub genes” should be changed to just “Hubgenes” or “Hub genes”

• Line 84: “10and 14” should be fixed to “10 and 14”

• Line 162: “wasnoted” should be broken up into “was noted”

• 172: “97eigengenes” and “showedsignificant” should be split into “97 eigengenes” and “showed significant”

• 195: “Hubgeneswith” should be split up

• 223: “B12can” should be split up

Response: Thank you for your comments. We have corrected all the typographical errors you mentioned and other errors in this article.

• Figure 2D should be retitled “Number of nodes in modules” or something along those lines. Numbers is currently misspelled as numbers.

Response: Thanks for your comment. We're sorry that the title is misspelled. The title is now adjusted to “Number of nodes in modules”.

Reviewer #2: The authors addressed my comments. However, there are still some typographical errors.

These errors are all two words written together and they can be corrected easily by using a spellchecker.

pp1 line 34 HubgenesHub genes

pp6 line 195 hubgeneswith

pp6 line 200 weresimultaneously

pp6 line 202 wereidentified

pp7 line 241 promotestumor

pp8 line 273 mayhave

Response: Thank you for your comments. We have corrected all the typographical errors you mentioned and other errors in this article.

---

## [Editor Report · Decision Letter 2]

3 Aug 2020

PONE-D-20-10567R2

Novel genes associate with folic acid-mediated metabolism in mouse: a bioinformatics study

PLOS ONE

Dear Dr. Hu,

Thank you for submitting your manuscript to PLOS ONE. After careful consideration, we feel that it has merit but does not fully meet PLOS ONE’s publication criteria as it currently stands. Therefore, we invite you to submit a revised version of the manuscript that addresses the points raised during the review process.

ACADEMIC EDITOR:

While the scientific content of the manuscript is now suitable, there are still many basic grammatical and language errors throughout the manuscript. If published in its current form, it would reflect poorly on the authors and the journal. The manuscript needs significant language editing to make it suitable for publication (perhaps working with a professional language editing service could help).

We look forward to receiving your revised manuscript.

Kind regards,

Jishnu Das, Ph.D.

Academic Editor

PLOS ONE

---

## [Author Response · Author response to Decision Letter 2]

24 Aug 2020

Dear Editor,

Thank you for your comments.

We have polished the manuscript as required. We have selected a third-party organization to polish the manuscript, and the polishing report has been uploaded.

Best regards

Hu

---

## [Editor Report · Decision Letter 3]

27 Aug 2020

Novel genes associated with folic acid-mediated metabolism in mouse: A bioinformatics study

PONE-D-20-10567R3

Dear Dr. Hu,

We’re pleased to inform you that your manuscript has been judged scientifically suitable for publication and will be formally accepted for publication once it meets all outstanding technical requirements.

Kind regards,

Jishnu Das, Ph.D.

Academic Editor

PLOS ONE
---

## [Editor Report · Acceptance letter]

2 Sep 2020

PONE-D-20-10567R3 

Novel genes associated with folic acid-mediated metabolism in mouse: A bioinformatics study 

Dear Dr. Hu:

I'm pleased to inform you that your manuscript has been deemed suitable for publication in PLOS ONE. Congratulations! Your manuscript is now with our production department. 

Kind regards, 

on behalf of

Dr. Jishnu Das 

Academic Editor

PLOS ONE